# Morphological Characteristics, Fruit Qualities and Evaluation of Reproductive Functions in Autotetraploid Satsuma Mandarin (*Citrus unshiu* Marcow.)

**Miki Sudo** [1,†], **Kiichi Yasuda** [2,†], **Masaki Yahata** [1,\*], **Mai Sato** [2], **Akiyoshi Tominaga** [1], **Hiroo Mukai** [1], **Gang Ma** [1], **Masaya Kato** [1] and **Hisato Kunitake** [3]

[1] Faculty of Agriculture, Shizuoka University, 836 Ohya, Suruga-ku, Shizuoka-shi 422-8529, Shizuoka, Japan; sudo.miki@shizuoka.ac.jp (M.S.); tominaga.akiyoshi@shizuoka.ac.jp (A.T.); mukai.hiroo@shizuoka.ac.jp (H.M.); ma.gang@shizuoka.ac.jp (G.M.); kato.masaya@shizuoka.ac.jp (M.K.)

[2] School of Agriculture, Tokai University, 9-1-1 Toroku, Higashi-ku, Kumamoto-shi 862-8652, Kumamoto, Japan; yk964422@tsc.u-tokai.ac.jp (K.Y.); 7bnv1113@mail.u-tokai.ac.jp (M.S.)

[3] Faculty of Agriculture, University of Miyazaki, 1-1 Gakuenkibanadai-nishi, Miyazaki-shi 889-2192, Miyazaki, Japan; hkuni@cc.miyazaki-u.ac.jp

\* Correspondence: yahata.masaki@shizuoka.ac.jp; Tel.: +81-54-641-9500

† These authors contributed equally to the article.

**Abstract:** The morphological characteristics and fruit quality of an autotetraploid plant selected from nucellar seedlings of Satsuma mandarin (*Citrus unshiu* Marcow.) were investigated. Additionally, in order to evaluate the reproductive potential of male and female gametes of the tetraploid Satsuma mandarin, reciprocal crosses with diploid cultivars were also carried out. The tetraploid had significantly thick and round leaves, as compared to those of the diploid Satsuma mandarin. The sizes of the flowers and pollen grains of the tetraploid were significantly larger than those of the diploid. Pollen fertility of tetraploid was high compared with that of the diploid. The tetraploid produced seedless fruits. The fruit weight of the tetraploid was equal to that of the diploid. Compared to the diploid fruits, the tetraploid fruit had less sugar contents and more organic acid contents. Although the tetraploid fruits showed similar traits to other *Citrus* tetraploids such as thick and hard peels, the tetraploid had a higher content of carotenoids in the flavedo than the diploid, and the rind color of the tetraploid was much better. In the reciprocal crosses between the tetraploid Satsuma mandarin and diploid cultivars, some seeds were obtained, and triploid progenies were obtained in all cross combinations.

**Keywords:** fertility restoration; interploid crossing; polyploid breeding; triploid

## 1. Introduction

Satsuma mandarin (*Citrus unshiu* Marcow.) is an economically important fruit crop in Japan. Satsuma mandarin fruits have high quality juice, are easy to peel and have many functional components such as ascorbic acid, β-cryptoxanthin and hesperidin [1–4]. Therefore, Satsuma mandarins are used not only for fresh consumption but also for juice, preserve, candy, syrup and oriental medicine. Furthermore, the Satsuma mandarin produces seedless fruits because of its genetic sterility [5]. Using this trait, the Satsuma mandarin contributes to production of many excellent cultivars such as the 'Kiyomi' tangor, the 'Shiranui' mandarin, the 'Harumi' mandarin and the 'Setoka' tangor in Japan [6]. The Satsuma mandarin is also an important breeding material. As for cultivation, Satsuma mandarin is suited to the Japanese climate, and show disease resistance and high productivity. In recent years, however, the environmental changes caused by global warming are negatively affecting the fruit quality of Satsuma mandarin causing such problems as coloring disorder, rind puffing and bland taste [7,8]. These have become major problems in the cultivation of the Satsuma mandarin in Japan.

The utilization of tetraploid plants in the genus *Citrus* and its related genera is very important in breeding to improve fruit quality and to increase tolerance to stresses. Triploid plants can produce seedless fruits which is a commercial trait [9–11], and tetraploid plants are important materials to breed triploid plants because triploid plants have been efficiently obtained by the reciprocal crosses between tetraploid and diploid plants [12–14]. Tetraploidization affects morphology and physiology, and is used to improve fruit quality [12,15,16]. In the genus *Fortunella*, tetraploid plants showed desirable traits for kumquats such as thicker pericarp and higher soluble solids content [12,15]. In regard to cultivation, citrus tetraploids have been shown to improve tolerance to environmental stresses such as salinity and drought [17–19].

We selected a tetraploid plant from a number of seedlings of Satsuma mandarin. This tetraploid showed vigorous growth and fortunately produced many flowers and fruits for the first time 7 years after it was top-graft onto trifoliate orange [*Poncirus trifoliata* (L.) Raf.] (Figure 1). As far as we know, the morphological characteristics, fruit quality and the reproductive potential of the tetraploid in Satsuma mandarin have not yet been reported. To obtain the basal information of the tetraploid plant of the Satsuma mandarin, in the present study, we investigated the morphological characteristics and fruit qualities. To use breeding materials, furthermore, we evaluated the reproductive potential of tetraploid Satsuma mandarin as a male or a female parent by crossing it with diploid cultivars.

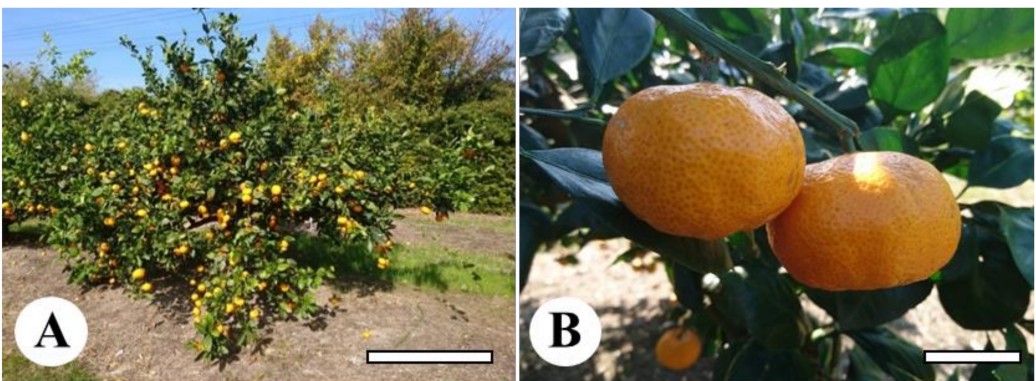

**Figure 1.** The tetraploid selected from nucellar seedlings of Satsuma mandarin. (**A**): Approximately 10 years after grafting onto trifoliate orange (Bar = 1 m). (**B**): Mature fruits (Bar = 3 cm).

## 2. Materials and Methods

### 2.1. Plant Materials

A tetraploid plant (tetraploid) selected from seedlings of 'Ishizuka-Wase' Satsuma mandarin (*Citrus unshiu* Marcow.) was used in the present study. A diploid 'Miyagawa-Wase' Satsuma mandarin (diploid) was used as control. These plant materials were grafted onto trifoliate orange [*Poncirus trifoliata* (L.) Raf.] and maintained for approximately 10 years in the Center for Education and Research in Field Science, Faculty of Agriculture, Shizuoka University. In SSR analysis, 5 species including the 'Okitsu-wase' Satsuma mandarin (*C. unshiu*), the Ponkan mandarin (*C. reticulata* Blanco), the Tachibana mandarin [*C. tachibana* (Makino) Tanaka], the Meiwa kumquat (*Fortunella classifolia* Swingle) and the Hongkong kumquat [*F. hindsii* (Champ. ex Benth.) Swingle] were used as comparative varieties. In the evaluation of the reproductive functions, furthermore, 2 diploid monoembryonic *Citrus* cultivars {the 'Banpeiyu' pummelo [*C. maxima* (Burm.) Merr.] and the 'Harehime' mandarin} were used for the reciprocal crosses.

### 2.2. Confirmation of the Origin of the Tetraploid Satsuma Mandarin by SSR Analysis

Total DNA was extracted from the young leaves of each plant following the method described by Doyle and Doyle [20] with some modifications. The isolated gDNA was diluted to 5 ng $\mu L^{-1}$ for SSR analysis. 6 SSR markers produced by Shimizu et al. [21] were used in the present study (Table S1). PCR amplifications and electrophoresis of

PCR-amplified products were carried out according to the methods described by Yasuda et al. [22]. PCR amplifications were performed in a thermal controller (Biometra TAdvanced Thermocycler 96G, Analytik Jena, Jena, Germany) in a 5 µL final volume containing 0.5 ng µL$^{-1}$ gDNA, 0.2 µM forward and reverse primers and 2.5 µL GoTaq Green Master Mix (Promega, Madison, WI, USA). The thermal cycling conditions were as follows: an initial denaturation step at 94 °C for 3 min. followed by 32 cycles at 94 °C for 20 s, a primer annealing step at 56 °C for 35 s, and a primer extension step at 72 °C for 35 s, with a final extraction step at 72 °C for 10 min. Poly-acrylamide gel electrophoresis of PCR-amplified products was carried out on a 12% polyacrylamide gel at 200 V for 80 min. The gel was stained with TBE buffer containing 0.1 µg mL$^{-1}$ ethidium bromide for 30 min. The separation of alleles was observed under UV irradiation using the stained gel.

### 2.3. Morphological Characteristics of Leaf, Flower and Pollen

Morphological characteristics of the leaves (leaf blade size, leaf weight per unit, guard cell size and stoma density), flowers (size of flower bud, petal, pistil and ovary, and number of petals, stamens and ovules per ovary) and pollen grains (size) were investigated in 2014. Full expanded leaves and flowers just before bloom were collected in the middle of August and the middle of May, respectively. The pollen grains were collected from flowers just before bloom of the middle of May. 20 samples were used for each measurement. Furthermore, pollen fertility was evaluated by stainability and in vitro germination. Pollen stainability was estimated by staining the samples with 1% acetocarmine after squashing nearly mature anthers on a glass slide. In vitro germination of the pollen grains was performed on microscope slides covered with a 2 mm layer of 1% (*w/v*) agar medium containing 10% sucrose. Five stamens, each from different flowers, were rubbed on the agar medium, and the slides were then incubated for 10 h in a moistened chamber at 25 °C in the dark. Each test was evaluated from 1000 grains with five repetitions. The significance of defferences was determined using the Student's *t*-test.

### 2.4. Evaluation of Fruit Quality

Fruit quality was evaluated for two years (2014 and 2015) although carotenoid quantification wasn't evaluated in 2014. Fruits from both the diploid and tetraploid Satsuma mandarin were harvest in the end of November.

20 fruits were used for measuring the fruit size, percentage of rind per a fruit, rind color, degree of rind puffing, number of seeds, soluble solids content (SSC) and titratable acidity (TA). Rind color was assessed color chart (FUJIHIRA INDUSTRY Co. Ltd., Tokyo, Japan). Degree of rind puffing was assessed according to the methods of Kawase and Hirai [23], and scored using a scale from 0 (non-puffy) to 3 (seriously puffy) by touch.

The sugar and organic acid contents of the juice were determined according to the method of Mukai et al. [24]. The sugar content was calculated with sucrose, fructose and glucose. As for organic acid content, citric and malic acids were measured. 10 fruits were used.

The identification and quantification of carotenoids in the juice and flavedo were carried out according to the methods described by Kato et al. [25] in 2015. All-*trans*-violaxanthin, 9-*cis*- violaxanthin, lutein and β-cryptoxanthin were quantified. The contents were expressed as µg g$^{-1}$ fresh weight. The carotenoid quantification was performed in three replications.

The significance of defferences was determined using the Student's *t*-test.

### 2.5. Crossing for Evaluation of the Reproductive Functions

The crossing test was conducted in 2014. The flowers were pollinated immediately after emasculation and covered with paraffin paper bags. Seeds were collected from each fruit of the crosses at maturity. The extracted seeds were then classified into 3 groups: developed, undeveloped and empty seeds. Developed seeds obtained from fruits of the 'Harehime' mandarin were then placed on moistened filter paper and maintained at 25 °C.

Other developed seeds and undeveloped seeds were cultured on Murashige and Skoog (MS) medium [26] containing 500 mg $L^{-1}$ malt extract, 30 g $L^{-1}$ sucrose, and 2 g $L^{-1}$ gellan gum at 25 °C under continuous illumination (38 μmol $m^{-2}$ $s^{-1}$). After germination, the seedlings were transplanted into pots containing vermiculite and transferred to a greenhouse. Ploidy analysis of the seedlings were performed by flow cytometry (FCM) (EPICS XL; Beckman Coulter, Fullerton, CA, USA) using young leaves according to the methods of Yahata et al. [27,28].

## 3. Results

### 3.1. Confirmation of the Origin of the Tetraploid Satsuma Mandarin by SSR Analysis

The gametic seedlings may have been mixed with the nucellar seedlings because the tetraploid plant used in the present study was selected from seeds of diploid Satsuma mandarin. Therefore, we needed to confirm the origin of the tetraploid. The genetic origin of the tetraploid was analyzed with 6 SSR markers. In NSX165 and GSR3138 out of 6 primers, specific bands were observed among each species (Figure 2). The tetraploid had the same banding patterns as diploid 'Miyagawa-wase' Satsuma mandarin. This result showed that this tetraploid plant was confirmed to be derived from the nucellar embryo of an original diploid, and to be an autotetraploid.

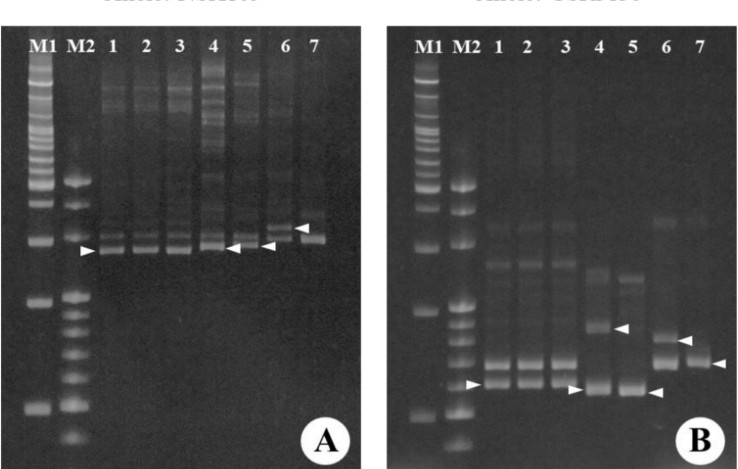

**Figure 2.** SSR analysis of NSX165 (**A**) and GSR3138 (**B**) allele in the tetraploid satsuma mandarin induced by colchicine treatment to nucellar embryos. M1: 100 bp ladder marker, M2: 20 bp ladder marker, 1: The diploid Satsuma mandarin ('Miyagawa-wase'), 2: The tetraploid Satsuma mandarin, 3: 'Okitsu-wase' Satsuma mandarin, 4: Ponkan mandarin, 5: Tachibana mandarin, 6: Meiwa kumquat, 7: Hongkong kumquat. Arrowheads indicate the bands specific to the cultivars or species.

### 3.2. Morphological Characteristics of Leaf, Flower and Pollen

The morphological characteristics of the tetraploid were compared with those of the diploid. The sizes of the leave and guard cells of the tetraploid were larger than those of the diploid (Table 1, Figure 3A). The flower organ of the tetraploid showed normal morphology (Figure 3B). The flower bud width and ovary of the tetraploid were also significantly larger than those of the diploid (Table 2, Figure 3B). The tetraploid had a significantly increased number of stamens per flower.

**Table 1.** Comparison of the morphological characteristics of the leaves of the diploid and tetraploid Satsuma mandarins.

| Strains | Leaf Blade (mm) | | Shape Index of Leaf Blade [2] | Guard Cell (µm) | | Stoma Density (No. mm$^{-2}$) |
| --- | --- | --- | --- | --- | --- | --- |
| | Length | Width | | Length | Width | |
| Diploid | 99.4 | 41.7 | 2.4 | 24.3 | 20.2 | 495.7 |
| Tetraploid | 109.9 | 58.6 | 1.9 | 27.9 | 26.1 | 319.3 |
| *t*-test | ** [1] | ** | ** | * | ** | ** |

[1] *, **: mean significantly different at 5, 1% levels by t-test, respectively. [2] Length of leaf blade/Width of leaf blade.

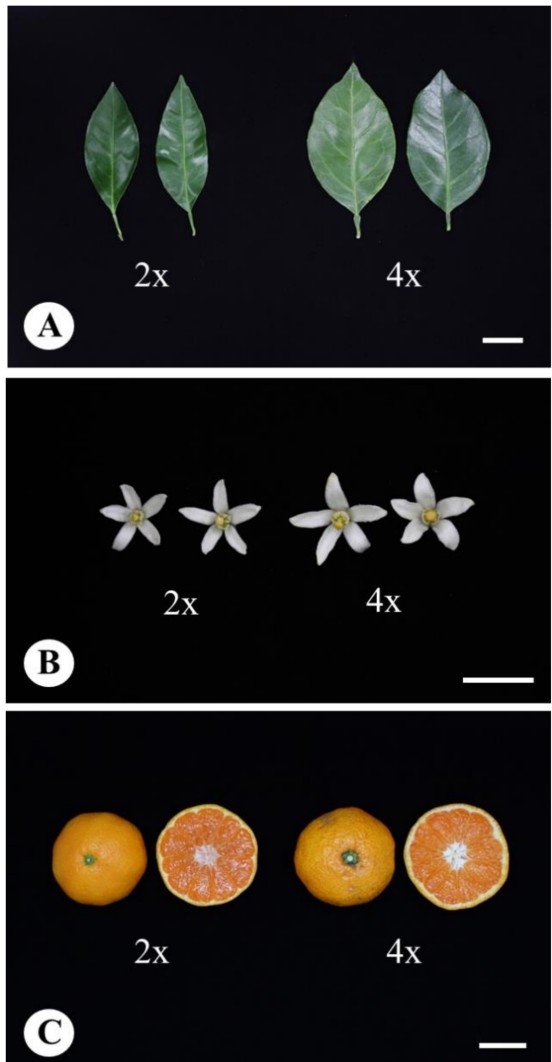

**Figure 3.** The morphological characteristics of leaves (**A**), flowers (**B**) and fruits (**C**) in diploid and tetraploid Satsuma mandarin. (Bar = 3 cm).

**Table 2.** Comparison of the morphological characteristics of the flowers of the diploid and tetraploid Satsuma mandarins.

| Strains | Flower Bud (mm) | | No. of Petals | Petal (mm) | | Pistil (mm) | Ovary (mm) | | No. of Stamens |
| --- | --- | --- | --- | --- | --- | --- | --- | --- | --- |
| | Length | Width | | Length | Width | | Height | Diameter | |
| Diploid | 19.0 | 7.5 | 5.0 | 19.0 | 7.8 | 19.7 | 5.1 | 3.7 | 14.5 |
| Tetraploid | 19.1 | 10.8 | 4.9 | 19.6 | 9.5 | 21.2 | 5.8 | 5.2 | 15.0 |
| *t*-test | NS [1] | ** | NS | NS | ** | ** | ** | ** | ** |

[1] NS: No significantly different, **: mean significantly different at 1% levels by *t*-test.

The average size of the pollen grains from the tetraploid was also larger than that of the grains from the diploid (Table 3, Figure 4A,B). SEM observations revealed that the shape of pollen grains in the diploid showed severely depressed morphology (Figure 4A), whereas a part of the pollen grains of the tetraploid was solid and elliptical in shape (Figure 4B); these pollen grains were thus presumed to be fertile. The pollen fertility of the tetraploid (75.7% stainability and 5.3% pollen germination rate) was significantly higher than that of the diploid (56.6% and 1.4%) (Table 3, Figure 4C,D).

**Table 3.** Comparison of the morphological characteristics and fertility of pollen grains of the diploid and tetraploid Satsuma mandarins.

| Strains | Pollen Grain (μm) | | Shape Index of Pollen Grain [2] | Fertility (%) | |
|---|---|---|---|---|---|
| | Length | Width | | Stainability | In Vitro Germination |
| Diploid | 29.7 | 23.4 | 1.3 | 56.6 | 1.4 |
| Tetraploid | 42.6 | 32.7 | 1.3 | 75.7 | 5.3 |
| *t*-test | ** [1] | ** | NS | * | ** |

[1] NS: No significantly different, *, **: mean significantly different at 5, 1% levels by *t*-test, respectively. [2] Length of pollen grain/Width of pollen grain.

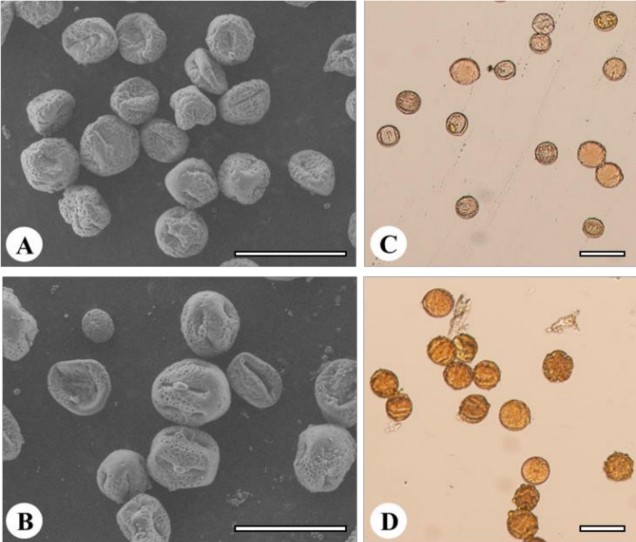

**Figure 4.** Scanning electron micrographs (**A,B**; Bars = 50 μm) and stainability by 1% acetocarmine (**C,D**; Bars = 50 μm) in pollen of diploid (**A,C**) and tetraploid (**B,D**) Satsuma mandarin.

### 3.3. Evaluation of Fruit Quality

The tetraploid had a rough rind and a flattened fruit. The fruit weight of the tetraploid was equal to that of the diploid (Table 4, Figures 1B and 2C). The percentage of rind weight per fruit in the tetraploid increased significantly in comparison with that of the diploid. The rind color of the tetraploid was much better than that of the diploid. There was no significant difference in rind puffing between the tetraploid and the diploid. The tetraploid showed low SSC and the sugar contents in comparison with that of diploid, while TA and the organic acid contents in the tetraploid was significantly higher than that of the diploid in both years (Table 5). As for carotenoid contents (Table 6), there was no significant difference between the tetraploid and diploid juice. On the other hand, the tetraploid had a higher total of carotenoid contents than the diploid in the flavedo, especially in the contents of All-*trans*-violaxanthin, 9-*cis*-violaxanthin and β-cryptoxanthin which resulted in a significantly deeper shade of orange. Moreover, the tetraploid fruits had no seed under field cultivation.

**Table 4.** Comparison of the morphological characteristics of fruits of the diploid and tetraploid Satsuma mandarins.

| Year | Strains | Fruit Weight (g) | Rind Per a Fruit (%) | Fruit (cm) | | Shape Index of Fruit [2] | Rind Color | Rind Puffing | No. of Seeds | Soluble Solid Content (%) | Titratable Acidity (%) | Soluble Solid-Acid Ratio [3] |
|---|---|---|---|---|---|---|---|---|---|---|---|---|
| | | | | Diameter | Height | | | | | | | |
| 2014 | Diploid | 126.5 | 18.5 | 67.6 | 52.6 | 1.3 | 7.7 | 0.2 | 0 | 13.6 | 0.8 | 17.0 |
| | Tetraploid | 126.7 | 26.5 | 71.2 | 48.7 | 1.5 | 8.6 | 0.4 | 0 | 10.4 | 1.0 | 10.4 |
| | *t*-test | NS [1] | ** | * | ** | ** | ** | NS | NS | ** | ** | ** |
| 2015 | Diploid | 124.8 | 21.8 | 67.2 | 53.5 | 1.3 | 7.7 | 0.2 | 0 | 11.8 | 0.8 | 14.8 |
| | Tetraploid | 128.3 | 28.5 | 70.0 | 50.8 | 1.4 | 8.6 | 0.4 | 0 | 10.4 | 1.0 | 10.4 |
| | *t*-test | NS | ** | * | * | ** | ** | NS | NS | ** | ** | ** |

[1] NS: No significantly different, *, **: mean significantly different at 5, 1% levels by *t*-test, respectively. [2] (Fruit diameter/Fruit height) × 100. [3] (Soluble solid content/Titratable acidity) × 100.

**Table 5.** Comparison of sugar and organic acid contents in the juice of the diploid and tetraploid Satsuma mandarins.

| Year | Strains | Sugar Content (%) | | | | Organic Acid Content (%) | | |
|---|---|---|---|---|---|---|---|---|
| | | Sucrose | Glucose | Fructose | Total | Citric Acid | Malic Acid | Total |
| 2014 | Diploid | 7.06 | 2.99 | 2.71 | 12.76 | 0.73 | 0.06 | 0.79 |
| | Tetraploid | 5.16 | 2.56 | 2.42 | 10.14 | 0.93 | 0.11 | 1.04 |
| | *t*-test | ** [1] | * | NS | ** | ** | * | * |
| 2015 | Diploid | 8.98 | 2.29 | 2.28 | 13.55 | 0.58 | 0.10 | 0.68 |
| | Tetraploid | 6.72 | 2.15 | 2.25 | 11.11 | 0.76 | 0.18 | 0.95 |
| | *t*-test | ** | NS | NS | ** | ** | ** | ** |

[1] NS: No significantly different, *, **: mean significantly different at 5, 1% levels by *t*-test, respectively.

**Table 6.** Comparison of the carotenoid contents in the juice and flavedos of the diploid and tetraploid Satsuma mandarins.

| Strains | Juice (µg g$^{-1}$) | | | | | Flavedo (µg g$^{-1}$) | | | | |
|---|---|---|---|---|---|---|---|---|---|---|
| | All-*trans*-Violaxanthin | 9-*cis*-Violaxanthin | Lutein | β-Cryptoxanthin | Total | All-*trans*-Violaxanthin | 9-*cis*-Violaxanthin | Lutein | β-Cryptoxanthin | Total |
| Diploid | 0.6 | 2.2 | 0.6 | 23.2 | 26.6 | 15.0 | 87.3 | 13.3 | 81.5 | 197.1 |
| Tetraploid | 0.5 | 2.7 | 0.2 | 20.6 | 24.0 | 29.5 | 184.2 | 13.6 | 113.1 | 340.3 |
| *t*-test | NS [1] | NS | NS | NS | NS | ** | ** | NS | * | ** |

[1] NS: No significantly different, *, **: mean significantly different at 5% and 1% levels by *t*-test.

### 3.4. Crossing for Evaluation of the Reproductive Functions

In order to evaluate the reproductive potential of the tetraploid Satsuma mandarin, reciprocal crosses with 2 diploid cultivars were carried out in the present study (Table 7). Developed seeds were obtained from the crosses with the tetraploid as the seed and pollen parent although a lot of undeveloped seeds were obtained when pollinated with the tetraploid as the pollen parent. The number of developed seeds per fruit in the crossing with the tetraploid was larger than that of the diploid. Developed and undeveloped seeds obtained from the reciprocal crosses between diploid cultivars and the tetraploid were germinated on moistened filter paper and MS medium, respectively. Consequently, 6 seedlings were obtained from the cross between the tetraploid and 'Banpeiyu' pummelo. In the cross between 'Harehime' mandarin and the tetraploid, 2 and 7 seedlings were grown from developed and undeveloped seeds, respectively. After being transplanted to soil, these seedlings grew normally (Figure 5A). Moreover, seeds of the tetraploid Satsuma mandarin were polyembryonic.

**Table 7.** Fruit sets and seed contents of the reciprocal crosses between the tetraploid Satsuma mandarin and diploid cultivars.

| Cross Combination | | No. of Flowers Pollinated | No. of Fruit Set | Fruit Set (%) | No. of Seeds | | | No. of Developed Seeds Per Fruit [1] |
|---|---|---|---|---|---|---|---|---|
| Seed Parent | Pollen Parent | | | | Developed | Undeveloped | Empty | |
| Diploid | Banpeiyu pummelo | 20 | 15 | 75.0 | 9 | 0 | 0 | 0.60 |
| Tetraploid | Banpeiyu pummelo | 10 | 4 | 40.0 | 3 | 0 | 0 | 0.75 |
| Harehime | Diploid | 20 | 8 | 40.0 | 1 | 4 | 0 | 0.12 |
| Harehime | Tetraploid | 20 | 12 | 60.0 | 15 | 53 | 0 | 1.25 |

[1] No. of developed seeds/No. of fruit set.

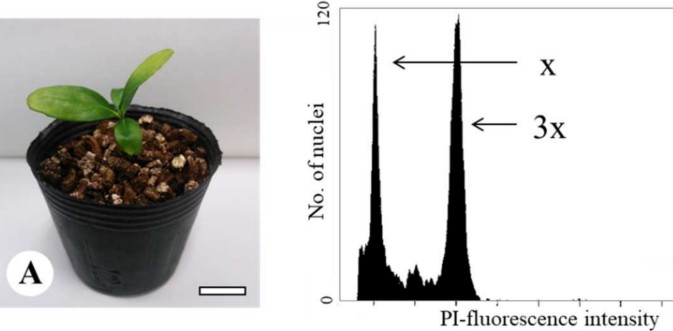

**Figure 5.** A seedling obtained from the cross between the tetraploid Satsuma mandarin and 'Banpeiyu' pummelo (**A**, Bar = 3 cm) and its flow cytometry analysis (**B**). x: Haploid pummelo as an internal standard, 3x: A triploid seedling obtained from the cross between the tetraploid Satsuma mandarin and 'Banpeiyu' pummelo.

These seedlings were analyzed to determine the ploidy level by flow cytometry (Table 8). 4 out of 6 seedlings showed the fluorescence intensity equal to that of the tetraploid, and the remaining 2 seedlings showed triploid DNA value in the cross between the tetraploid and 'Banpeiyu' pummelo (Figure 5B). On the other hand, all of the seedlings obtained from developed seeds were tetraploid in the cross between 'Harehime' mandarin and the tetraploid, whereas the seedlings derided from the undeveloped seeds were all triploids.

**Table 8.** Evaluation of the ploidy level of seedlings obtained from the reciprocal crosses between the tetraploid Satsuma mandarin and diploid cultivars.

| Cross Combination | | Kind of Seed | No. of Seedlings Examined | Ploidy Level | | |
|---|---|---|---|---|---|---|
| Seed Parent | Pollen Parent | | | 2X | 3X | 4X |
| Diploid | Banpeiyu pummelo | Developed | 20 | 20 | 0 | 0 |
| Tetraploid | Banpeiyu pummelo | Developed | 6 | 0 | 2 | 4 |
| Harehime | Diploid | Developed | 1 | 1 | 0 | 0 |
| | | Undeveloped | 1 | 1 | 0 | 0 |
| Harehime | Tetraploid | Developed | 2 | 0 | 0 | 2 |
| | | Undeveloped | 7 | 0 | 7 | 0 |

## 4. Discussion

As reported in genus *Citrus* and its related genera, tetraploid plants showed round and thick leaves, large guard cell, flowers and pollen grains, and poor fruit characteristics such as rough and thick rind and high organic acid as compared with diploid plants [12,16,29]. The tetraploid Satsuma mandarin investigated in the present study showed mostly the typical morphological characteristics and fruit quality of tetraploid *Citrus* and its related plants. But this tetraploid Satsuma mandarin had desirable traits for storage such as thick rind and juice with high TA and organic acid contents, so we should carry out the storage research about this tetraploid Satsuma mandarin in the future.

The coloration of the rind of the tetraploid Satsuma mandarin was better than that of the diploid. The color of rind and juice in *Citrus* spp. is derived from carotenoids [1]. Especially, rind of Satsuma mandarin is a type of containing a lot of β-cryptoxanthin and violaxanthin showing orange color [25,30]. Our carotenoid measurements showed that tetraploid value of β-cryptoxanthin and violaxanthin was significantly higher than diploid one, resulting in the superior coloration in the rinds of the tetraploid Satsuma mandarin fruits.

The rinds of Satsuma mandarin fruits also contained a lot of flavonoids such as hesperidin and narirutin [31,32]. In East Asia, the dried rinds of mandarins such as the Satsuma mandarin and the Ponkan mandarin are used as an oriental medicine, and is called "Chinpi". Chinpi has been used as an aromatic stomachic, cold medicine, expectorant and antitussive [33,34]. In the previous study, polyploidization can influence the physiological and biochemical processes of the plant, and there are reports of increasing secondary metabolites of chromosome-doubling individuals in some plant species [35–38]. Kim et al. [37] reported that autotetraploid plants of *Cnidium officinale* Makino, which had been induced by treating in vitro shoots with oryzalin, had more gentisic acid, salicylic acid and naringin contents compared with those of the diploids, and the phenolic content of the tetraploids was significantly higher than that of the diploids. Tetraploid plants induced by colchicine treatment to shoot apical meristems of diploid seedlings in *Cichorium intybus* L. had more total phenolic compound and chlorogenic acid concentrations in their leaves than those of diploid plants [39]. In the present study, our results from the tetraploid Satsuma mandarin show an increase in the amount of the secondary metabolite carotenoid, and other secondary metabolites may also be affected by tetraploidization. We plan to measure the contents of flavonoids such as hesperidin and narirutin in the rind of this tetraploid.

Satsuma mandarin used in the present study is also known to have considerable female and male sterility [40], and the cause of the sterility is different from the previously mentioned plants in terms of genetic background. The female sterility of the Satsuma mandarin is due to degeneration of embryo-sac mother cells or embryo sacs [41], and the cause of male sterility is poor development of the anthers [5,42]. In particular, the male sterility of the Satsuma mandarin is considered to be a genetic-cytoplasmic male sterility type because the progeny of the Satsuma mandarin inherits the sterility [5,43,44]. In order to obtain detailed knowledge of male sterility of the Satsuma mandarin, Yang and Nakagawa [42] performed cytological observations of anther development and male gametogenesis, and

they revealed that the degeneration on microspores in the Satsuma mandarin is caused by an insufficient supply of essential nutrients, ascribed to the abnormal behavior of the tapetum. How the tetraploid Satsuma mandarin recovers pollen fertility is not clear at the present stage, but it is thought to be due to the tetraploidization suppressing the abnormal behavior of tapetum and improving the supply of essential nutrients to microspores. Detailed cytological observation of the tapetum during male gametogenesis, is needed in the future research to determine how the tetraploid Satsuma mandarin recovers fertility.

In the reciprocal crosses between the tetraploid Satsuma mandarin and diploid cultivars, both triploid and tetraploid progenies were appeared in each cross combination. When tetraploid Satsuma mandarin was pollinated with pollen from a 'Banpeiyu' pummelo, more than one seedling germinated from a seed. Because the tetraploid Satsuma mandarin had polyembryonic seeds, tetraploid progenies were considered as the origin of the nucellar embryos. On the other hand, 2 developed seed-derived tetraploids were obtained from the cross between the 'Harehime' mandarin and the tetraploid. The appearance of tetraploid progenies in 2x × 4x was reported by some previous studies [13,14,45]. Like these reports, it presumed that the unexpected tetraploid originated from the fertilization between the diploid female gamete produced by chromosome doubling of the unreduced gamete and the diploid reduced male gamete.

## 5. Conclusions

The tetraploid Satsuma mandarin showed a similar morphology to other tetraploids in genus *Citrus* and its related genera, such as large leaves, flowers and pollen grains. Tetraploid fruits showed lower sugar contents and higher acidity than those of the diploid, but their rind was well colored and had high carotenoid contents. Furthermore, when the tetraploid was used as the seed and pollen parents, fully developed seeds and triploid progenies were obtained from both crosses. This result showed that the tetraploid Satsuma mandarin can be used as breeding material. In the future, we plan to carry out research that not only utilizes triploid breeding but also investigates the potential of this tetraploid Satsuma mandarin for commercial growing.

**Supplementary Materials:** The following are available online at https://www.mdpi.com/article/10.3390/agronomy11122441/s1, Table S1: Information on SSR markers used in the present study with their GenBank accession numbers, linkage groups, primer sequences and bibliographic references.

**Author Contributions:** Conceptualization, M.S. (Miki Sudo), K.Y. and M.Y.; methodology, M.S. (Miki Sudo), K.Y., M.Y., H.M. and M.K.; formal analysis, M.S. (Miki Sudo), K.Y., M.Y., A.T., G.M. and M.K.; investigation, M.S. (Miki Sudo), K.Y., M.Y., M.S. (Mai Sato), A.T., H.M., G.M. and M.K.; resources, M.S. (Miki Sudo), K.Y., M.Y. and H.K.; data curation, M.S. (Miki Sudo), K.Y., M.Y., M.S. (Mai Sato), A.T., G.M. and M.K.; writing—original draft preparation, M.S. (Miki Sudo) and K.Y.; writing—review and editing, M.Y., M.S. (Mai Sato), A.T., H.M., G.M., M.K. and H.K.; visualization, M.S. (Miki Sudo), K.Y. and M.Y.; supervision, M.Y.; project administration, M.Y. All authors have read and agreed to the published version of the manuscript.

**Funding:** This research received no external funding.

**Data Availability Statement:** All data were presented in this paper.

**Conflicts of Interest:** The authors declare no conflict of interest.

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
