# Peer review of "Morphological Characteristics, Fruit Qualities and Evaluation of Reproductive Functions in Autotetraploid Satsuma Mandarin (Citrus unshiu Marcow.)"

_agronomy, doi:10.3390/agronomy11122441_

Round 1

Reviewer 1 Report

Nice presentation to discribe the morphological and fruit quality evaluation of 4X Satsuma mnadarin. Thou there already have many reports on 4X in citrus, but not yet on Satsuma mandarin, a typical male sterile and a leading citrus cultivar type in Japan and China. I ony haver some minor comments.

  1. The concrete cultivar name to derive the 4X type should be indicated. Since there have many Satsuma mandarin cultivars, simply mention it was exploited from Satsuma mandarin is not enough.
  2. There have too many tedious tables, some can be merged and some others can be put as supplementary tables.
  3. The text is generally too long, the wording needs to be more concise, and some routine description in introduction, M & M, results and dicussion can be simplified.

Reviewer 2 Report

Satsuma mandarin is an economically important fruit in East Asian countries. Tetraploid breeding is a powerful way to improve fruit quality and stress tolerance of Citrus. This manuscript analyzes the variations of autotetraploid Satsuma mandarin on morphological characteristics, fruit qualities and reproductive potential of gametes. This investigation is valuable for understanding of variation of autotetraploidization in Citrus.

Reviewer 3 Report

In the manuscript “  Morphological Characteristics, Fruit Qualities and Evaluation of Reproductive Functions in Autotetraploid Satsuma Mandarin (Citrus unshiu Marcow.)« authors  Miki Sudo, Kiichi Yasuda, Masaki Yahatar, Mai Sato, Akiyoshi Tominaga, Hiroo Mukai, Gang Ma, Masaya Kato and Hisato Kunitake  evaluated the reproductive potential of tetraploid Satsuma mandarin as a male or a female parent by crossing it with diploid cultivars.

Abstract

OK

Key words

OK

Introduction

Is informative, concise, with the appropriate literature cited.

Materials and Methods

What statistics was performed?

Results

OK

Discussion

L 297 thepreviously  write space between the and previously

Conclusions

L 332-333 In the following statement, the English is not OK: This result showed that the tetraploid Satsuma mandarin has no problem in the reproductive potential of female and male gametes.

Special comments

The idea of MS is interesting and the results are clearly presented. I want to highlight the clear and comprehensive work of the authors. I just want to know, which statistic was used. The methods employed are explained, and the results are consistent with the purposes.

My suggestions: minor revision
